# Klotho an Autophagy Stimulator as a Potential Therapeutic Target for Alzheimer’s Disease: A Review

**DOI:** 10.3390/biomedicines10030705

**Published:** 2022-03-18

**Authors:** Tsz Yan Fung, Ashok Iyaswamy, Sravan G. Sreenivasmurthy, Senthilkumar Krishnamoorthi, Xin-Jie Guan, Zhou Zhu, Cheng-Fu Su, Jia Liu, Yuxuan Kan, Yuan Zhang, Hoi Leong Xavier Wong, Min Li

**Affiliations:** 1Mr. & Mrs. Ko Chi-Ming Centre for Parkinson’s Disease Research, School of Chinese Medicine, Hong Kong Baptist University, Hong Kong, China; 16221044@life.hkbu.edu.hk (T.Y.F.); sravangs@hkbu.edu.hk (S.G.S.); senthilkumark@hkbu.edu.hk (S.K.); 21481555@life.hkbu.edu.hk (X.-J.G.); zhuzhou1@hkbu.edu.hk (Z.Z.); 20481403@life.hkbu.edu.hk (C.-F.S.); 20482248@life.hkbu.edu.hk (J.L.); yuxuan@hkbu.edu.hk (Y.K.); 2Institute for Research and Continuing Education, Hong Kong Baptist University, Shenzhen 518057, China; 3Centre for Trans-Disciplinary Research, Department of Pharmacology, Saveetha Dental College and Hospitals, Chennai 600077, Tamil Nadu, India; 4Shenzhen Key Laboratory of Neurosurgery, Department of Neurosurgery, Shenzhen Second People’s Hospital, The First Affiliated Hospital of Shenzhen University, Shenzhen 518025, China; zhangyuan2019@email.szu.edu.cn; 5School of Chinese Medicine, Hong Kong Baptist University, Hong Kong, China

**Keywords:** Klotho, Alzheimer’s disease, autophagy, neurodegenerative disease, autophagy lysosomal pathway (ALP)

## Abstract

Alzheimer’s disease (AD) is an age-associated neurodegenerative disease; it is the most common cause of senile dementia. Klotho, a single-pass transmembrane protein primarily generated in the brain and kidney, is active in a variety of metabolic pathways involved in controlling neurodegeneration and ageing. Recently, many studies have found that the upregulation of Klotho can improve pathological cognitive deficits in an AD mice model and have demonstrated that Klotho plays a role in the induction of autophagy, a major contributing factor for AD. Despite the close association between Klotho and neurodegenerative diseases, such as AD, the underlying mechanism by which Klotho contributes to AD remains poorly understood. In this paper, we will introduce the expression, location and structure of Klotho and its biological functions. Specifically, this review is devoted to the correlation of Klotho protein and the AD phenotype, such as the effect of Klotho in upregulating the amyloid-beta clearance and in inducing autophagy for the clearance of toxic proteins, by regulating the autophagy lysosomal pathway (ALP). In summary, the results of multiple studies point out that targeting Klotho would be a potential therapeutic strategy in AD treatment.

## 1. Introduction

Alzheimer’s disease (AD) is the most common neurodegenerative disease and the most frequent cause of dementia worldwide [1,2]. According to the World Health Organization, the prevalence of AD has been increasing. It is estimated that over 50 million people in the world are afflicted with AD and the related form of senile dementia [3,4]. AD is initially associated with memory loss, but as the disease progresses, people with AD develop cognitive dysfunction, impulsive or unpredictable behavioral problems and abnormal personality changes [5,6,7,8,9]. Neurologically, AD is characterized by neuroinflammation, extracellular amyloid-beta (Aβ) plaque, and intracellular neurofibrillary tangle (NFT) deposition which are associated with a progressive cognitive decline and neurodegeneration [10,11,12,13,14,15]. Over the past 20 years, the amyloid cascade hypothesis and tau hypothesis have been the two most popular hypotheses used to explain AD pathogenesis [16,17,18]. The amyloid cascade hypothesis proposes that the excessive generation of Aβ plaques is derived from the cleavage of the amyloid precursor protein (APP) deposit in the hippocampus and basal segment, causing neuronal damage [18,19,20]. The tau hyperphosphorylation hypothesis proposes that the deposition of intracellular NFTs formed by hyperphosphorylated tau protein causes mitochondrial dysfunction, synaptic deficits and neuronal death, eventually leading to neurodegeneration and cognitive decline [21,22,23]. Apart from these two hypotheses, researchers are also exploring new theories to explain the etiology and pathogenesis of AD. These new theories include, for example, prion transmission, gamma oscillations, lysosome dysfunction, and calcium dysregulation. It is hoped that they will be able to better explain the pathological mechanism of AD, ultimately leading to a safe, and effective therapy for AD [6,7,13].

New evidence has raised the possibility that Klotho may be an anti-AD target protein. Klotho is a protein encoded by the *Klotho* gene, which is highly expressed in the kidney and the choroid plexus in the brain [24,25,26,27,28,29]. Previous studies showed that Klotho protein might take part in controlling oxidative stress, ER stress, Golgi apparatus stress, cell proliferation, apoptosis and autophagy [30,31,32,33]. Klotho is reported to be involved in numerous ageing-associated pathologies [34,35], such as cardiovascular disease, chronic kidney disease, cancer, and neurodegenerative disease [36,37]. Evidence has also demonstrated that Klotho expression levels are reduced in ageing brains, as well as in the brains of patients in the early stage of AD [37,38,39]. Our understanding of the Klotho protein and its function related to the progression of AD is far from complete. However, studies with an aged amyloidogenic mice model have demonstrated that the overexpression of Klotho protein in the brain can improve AD-like pathology and cognitive impairment, and reverse neuronal damage. It can also ameliorate Aβ accumulation in a mice model by regulating the Aβ-related transporters and microglia transformation (Table 1). In the early stages of AD, degradative pathways namely autophagy, a ubiquitin-proteasome system (UPS), and chaperone-mediated autophagy (CMA) are impaired [40]. More importantly, intracellular accumulation of Aβ causes dysfunction in the lysosome and autophagy lysosomal pathway (ALP), thereby leading to neuronal loss [41]. Recent studies have demonstrated that the expression of Klotho and autophagy are related in AD pathology. Klotho over-expression can promote ALP in AD via activating the beclin1 pathway [42].

There is further evidence that Klotho expression promotes autophagy [44]. Autophagy is an essential pathway to maintain cellular homeostasis in eukaryotes [9,16]. It helps in degrading and removing a wide variety of substrates in the cells, from the senescence-related proteins to damaged organelles and even bacteria [37,44,45,46,47,48]. Autophagy is particularly important in neuronal cells because, as they age, they tend to accumulate intracellular toxicants and damaged organelles, which must be removed by autophagy [31,49]. Defects in autophagy have been shown to contribute to the pathogenic process of different diseases, including neurodegenerative diseases [50]. Several studies have shown a close relationship between autophagy and AD [51]. Autophagy is a key regulator in Aβ homeostasis [52,53]. It triggers the clearance of APP and APP cleavage products, for example, amyloid precursor protein-cleaved C-terminal fragments (APP-CTFs) and Aβ aggregates [54]. It is known that autophagy can remove Aβ plaques in the microglia through the autophagy receptor optineurin. In addition to inhibiting Aβ formation, the autophagy-lysosome pathway (ALP) can also suppress the formation of tau-enriched neurofibrillary tangles [54,55]. During AD progression, the continuous production of Aβ causes dysfunctional ALP [49]. For instance, Aβ affects the autophagolysosome maturation by interfering with the fusion of autophagosome and lysosome [49]. At the same time, Aβ blocks the retrograde movement of autophagolysosome toward the neuronal cell-body, which contributes to the accretion of the autophagic vacuoles in the brain [56]. As far as we know now, amyloid plaques and tau neurofibrillary tangles are still the main neuropathological changes in the AD patients’ brains [21]. A recent study has illustrated the co-relation between Aβ plaques/tau protein aggregation and autophagy, in AD models [54]. Autophagy malfunction is likely to facilitate the formation of Aβ and NFTs, resulting in the progression and aggravation of AD [56,57,58]. Thus, autophagy may take part in AD, and the autophagy mechanism may be a viable therapeutic target for treating AD [54,59]. Additionally, other studies have shown that Klotho protein can stimulate autophagy induction, leading to the facilitation of Aβ and NTFs clearance [60,61,62]. Therefore, Klotho protein might be a promising therapeutic target for treating AD. That is, it could undermine the development of AD-induced neuropathies via ameliorating Aβ and tau pathologies (Figure 1), at the same time inducing autophagy to protect neurons. Herein, we review the expression, location, structure, and the biological functions of Klotho; in particular, we include studies examining Klothos’ role in regulating Aβ clearance and autophagy activation.

## 2. Expression, Structure and Function of Klotho Protein

The anti-ageing gene *Klotho* was discovered in 1997 by Kuro-o and his colleagues during their attempt to develop transgenic mice that overexpress rabbit type I sodium-proton exchanger (NHE-1). In the study, they produced transgenic mice by microinjection, and then mated the mice in order to obtain homozygous mice for examining phenotypic mutations. One of these mice exhibited an ageing-like phenotype and an overall shorter life span. This insertional mutation is now referred to as Klotho [63,64]. In Greek mythology, Klotho (Clotho) was the youngest daughter of Zeus and she spun the thread of life from her distaff onto her spindle, hence the name was given to the gene and protein, related to longevity.

The *Klotho* gene is located on chromosome 13q12. It is 50 kb long and consists of 5 exons and 4 introns with a molecular weight of about 130 kDa [65,66,67,68,69,70]. Both mouse and human *Klotho* gene produce two types of Klotho, the transmembrane form and the secreted form, through the alternative RNA splicing and ectodomain shedding of the membrane-bound Klotho by cell-surface protease (sheddases) [71,72]. *Klotho* gene-encoded type 1 single-pass transmembrane glycoprotein is situated in the cell membrane and Golgi apparatus. Of note, the internal domain is extremely short [72]. Lacking a functional domain, it consists of a larger extracellular domain with two internal repeats, KL1 and KL2 [29]. These repeats are cleaved by the sheddases, such as Beta-Secretase 1 (BACE1), A Disintegrin and Metalloproteinase Domain 10 (ADAM-10) and A Disintegrin and Metalloproteinase Domain 17 (ADAM-17), eventually generating a soluble form of Klotho which is released into the blood, cerebrospinal fluid and urine [65,73,74]. Loss of function in sheddases such as ADAM-17, and alpha-secretase leads to the accumulation of Aβ and a decrease in sheddases has been previously reported in AD [75,76]. Klotho is primarily found in the kidney and brain, but is also found in the skeletal muscle, lung, ovaries, urinary and testes [42,66]. Three types of Klotho protein have been distinguished: membrane, intracellular and secreted [66].

The membrane-bound Klotho plays a key role as a co-receptor, creating a complex with diverse fibroblast growth factor receptors (FGFRs). It has a selective ability to bind FGFRs to endocrine FGFs and to cooperate in their biological function [28,70]. For example, Klotho forms complexes with different FGFRs (FGFR1, FGFR3 and FGFR4), and has an increased affinity towards FGF23 [77,78]. FGF23 is a bone-derived phosphaturic hormone that helps in the control of energy and mineral metabolism; for example, it regulates the secretion of phosphate and the synthesis of vitamin D [26,69,79]. Most relevant here is that FGF23 acts on the Klotho-FGFRs complex and affects ageing [78,80]. The FGF23-Klotho system governs ageing through phosphate homeostasis regulation [78,81]. Emerging evidences demonstrate that premature ageing caused by the Klotho deficiency is related to an unbalanced phosphate metabolism [60]. Mice lacking FGF23 or Klotho exhibit the same phenotype, including hyperphosphatemia, hypervitaminosis D and premature ageing [80]. The activation of Klotho-FGFR complexes by FGF23 in the kidney helps reduce the phosphate level and consequently rescues the ageing-phenotype in FGF23-deficient or Klotho-deficient mice [80]. To be specific, FGF23 promotes the lowering of the sodium-dependent phosphate-transporter expression in the proximal tubules, to limit renal phosphate reabsorption [79,82]. Higher soluble αKlotho and a lower serum concentration of FGF23 has been reported to associate in older people and dementia patients [83]. FGF23 and αKlotho regulate the AD-induced neuroinflammation via the Wnt/β-catenin pathway [84].

FGF23 also plays a critical role in calcium homeostasis in the kidney. It activates calcium reabsorption and regulates calcium homeostasis through the transient receptor potential vanilloid-5 channel (TRPV5) in the renal distal tubules [26,36,69]. It is also known that Klotho can suppress inorganic phosphate reuptake in the renal proximal tubules via inhibiting the sodium-dependent phosphate co-transporter type-IIa (NaPi-IIa). Moreover, it downregulates the expression of 25-hydroxyvitamin D_3_ 1α-hydroxylase (CYP27B1), the main enzyme responsible for the synthesis of active calcitriol, which stimulates intestinal and renal inorganic phosphate and calcium absorption [36,69,70].

The secreted form of Klotho has anti-ageing, organ protecting, neuroprotective, and anti-fibrosis functions [85]. It is known that the expression of Klotho decreases with age in mice and humans. Studies provide evidence that Klotho deficiency stimulates ageing and shortens lifespans, while Klotho overexpression prolongs the lifespan in organisms, including *Caenorhabditis elegans*, Drosophila, and mice [42,86,87]. Secreted Klotho can regulate oxidative stress and inflammation through inhibiting multiple signaling pathways of cytokines and growth factors, such as insulin, insulin-like growth factor 1 (IGF-1), transforming growth factor β (TGF-β), and Interferon γ (IFNγ) [33,69,70]. *In vitro* studies have demonstrated that the overexpression of Klotho can suppress autophosphorylation of insulin and the IGF-1 receptor, and decreases oxidative stress. Several research findings demonstrated that suppressing insulin and IGF-1 signaling increases the lifespan in animals [88,89]. The anti-ageing function of Klotho is mainly because of its inhibitory effect on insulin/IGF-1 signaling [67,79]. Klotho has a marked effect in suppressing tyrosine phosphorylation of insulin and the IGF-1 receptor, which leads to the reduction of the activity of insulin receptor substrates (IRS) 1 and 2, and their association with phosphatidylinositol 3-kinase (PI3K), eventually suppressing the insulin/IGF-1 signaling pathways. Overall, the Klotho protein appears to induce insulin/IGF-1 resistance and inhibit the activation of the insulin/IGF-1 receptor, thereby blocking the downstream signaling cascade, and thus prolonging the lifespan [67,87,88]. Additionally, Klotho has anti-inflammatory properties. It can reduce the expression of the pro-inflammatory cytokines, IFNγ and Tumor Necrosis Factor α (TNFα). IFNγ is a cytokine with pleiotropic roles in immune and inflammatory responses, tissue homeostasis and disease immunosurveillance [90,91]. TNFα is known as a ‘master regulator’ of the production of pro-inflammatory cytokines to stimulate inflammation [92]. It is known that high levels of IFNγ and TNFα establish a pro-inflammatory environment in the cell that enhances the inflammatory response. In contrast, Klotho stimulates anti-inflammatory cytokine production. To be specific, Klotho can stimulate the production of interleukin 10 (IL-10), which is responsible for inhibiting the expression of pro-inflammatory cytokines, such as TNFα, thereby attenuating an inflammatory response [93]. According to earlier evidence, Klotho-deficient mice show an augmentation of the IFNγ and TNFα signaling pathways [26,60,65]. The secreted Klotho (sKlotho) is also an endogenous anti-fibrosis protein derived from kidneys. Earlier studies in animal models suggests that a decrease in Klotho expression can enhance/worsen fibrosis in organs, including the heart, kidney, lungs, skin, etc. Meanwhile, the overexpression of Klotho can reverse fibrosis of the heart and kidney, which is mediated by sKlotho [73,80]. Thus, Klotho may function as an anti-ageing, organ-protective, neuroprotective, and an anti-fibrosis protein through regulating signaling of different growth factors and cytokines [73].

## 3. Klotho Inhibits Neuroinflammation, Promotes Aβ Clearance, and Mitigates Tau Pathology in Alzheimer’s Disease

### 3.1. Klotho Inhibits the NLRP3/Caspase-1 Pathway

Neuroinflammatory processes have been shown to participate in AD development and progression. In the immunological context, neuroinflammation associated with AD consists of increased secretion of pro-inflammatory cytokines, a reduction of regulatory T lymphocyte (Treg) activity, and the downregulation of immunological tolerance [94]. The key event in these neuroinflammatory processes is the activation of inflammasomes [23,95]. Inflammasomes are multiprotein complexes that not only provide a platform for inflammatory caspase recruitment, cleavage and activation, but also mediate pro-inflammatory cytokine secretion and maturation [96]. NLRP3 is one of the best-characterized inflammasome components that play a vital role in AD-associated neuroinflammation [96,97]. The aggregation of Aβ stimulates cerebral inflammation by activating microglia in AD mice model [98]. Toxic Aβ peptide is a well-known NLRP3 activator. NLRP3 inflammasome is a multimeric protein made up of NLRP3, an apoptosis-associated speck-like protein containing a caspase recruitment domain (ASC) and cleaved caspase-1. NLRP3 is stimulated by Aβ deposition inside the microglia, where it stimulates the cleavage and activity of a cysteine protease, caspase-1. Activated caspase-1 is known as the main IL-1β converting enzyme that stimulates the downstream pro-inflammatory cytokines IL-1β secretion [10,25,96,99]. High production of IL-1β is associated with increasing AD pathogenesis. Researchers have detected an increased level of IL-1β in the cortex and hippocampus of AD patients [98]. IL-1β is called the ‘master regulator’ in the brain inflammatory cascade because it controls the expression of other pro-inflammatory cytokines, such as TNF-α [99,100]. Induction of IL-1β expression initiates the inflammatory signaling pathway leading to cell death, neuronal injury and neuroinflammation. IL-1β regulates the production of APP and further increases the Aβ burden and plaque deposition, eventually leading to the development of AD [100]. Conversely, it has been shown that an NLRP3 deficiency can suppress brain caspase-1 and IL-1β activation, which remarkably inhibits amyloidosis and neuropathology, and ultimately enhances cognitive function in the AD mice model. These results support an important role for the NLRP3/caspase-1 pathway in the pathogenesis of AD [23,94,95]. With regard to Klotho, current findings show that Klotho overexpression can remarkably inhibit NLRP3/caspase-1 signaling pathway activation. A high concentration of NLRP3, ASC and active caspase-1, has been shown in APP/PS1 transgenic mice [35,97]. However, these increased expressions were reversed in transgenic mice that overexpress Klotho [35]. Current findings have demonstrated that, in addition to Aβ aggregation, the activation of NLRP3 inflammasome is related to tau hyperphosphorylation. Activated NLRP3 can induce tau hyperphosphorylation and aggregation [10,23,96]. An elevated Klotho level can significantly decrease tau hyperphosphorylation in AD-transgenic mice [35]. Thus, a high expression of Klotho protein can effectively suppress the activation of the NLRP3/caspase-1 pathway (Figure 2), and thereby, in turn, reduce IL-1β secretion [35,36]. These findings explain how the inhibition of inflammatory events, particularly the reduction of Aβ deposition, can prevent neuronal cell damage and cognitive decline, and possibly protect against AD development.

### 3.2. Klotho Promotes the Transformation of M1 Microglia to M2 Microglia

Microglia activation and accumulation in the brain, concentrated around the Aβ plaques, are prominent characteristics of AD. This suggests an involvement of microglia in the pathogenesis of AD [101,102]. Microglia are the resident myeloid cells that participate in maintaining neuronal connectivity and regulatory processes, both of which are critical for the development of the central nervous system (CNS). Microglia also participate in regulating cognitive functions and maintaining homeostasis by eliminating dying cells, misfolded protein and cellular debris [1,101,102]. Morphologically, microglia are classified as either resting or activated. The transformation of microglia from resting to the activated state is promoted by different extracellular cytokines or factors. There is evidence that activated microglia can be hazardous to neurons. For example, they regulate the engulfment of neuronal synapses, secrete inflammatory factors and exacerbate tau pathology [101,102,103]. Activated microglia have two distinct phenotypes: pro-inflammatory (M1) and anti-inflammatory (M2). M1 microglia release pro-inflammatory cytokines and cytotoxic substances that stimulate inflammatory responses, leading to neurotoxicity, Aβ deposition and neurodegeneration (Figure 3). In contrast, M2 microglia have a neuroprotective role in the CNS. They secrete trophic factors and anti-inflammatory cytokines that induce phagocytosis, increase the uptake and clearance of extracellular Aβ, and downregulate inflammation, thereby preserving neurons and protecting against cognitive dysfunction [101,102,103,104,105]. Of note, several studies have demonstrated that Aβ aggregation activates NLRP3 inflammasome, resulting in a high level of M1 microglia, microglial production of IL-1β and pro-inflammatory cytokines, eventually exacerbating AD pathogenesis [101,103,104]. The overexpression of Klotho has been reported to suppress neuroinflammation by inhibiting the NLRP3/caspase-1 pathway in amyloidosis mouse models [35,106]. In vivo studies have shown that the overexpression of Klotho in mouse models significantly inhibits NLRP3 and decreases the production of caspase-1 and IL-1β, subsequently promoting microglial differentiation to the anti-inflammatory M2 phenotype [35,106]. This scenario is consistent with the fact that Klotho overexpression can promote microglia differentiation in the brain. Klotho suppresses NLRP3 activation and stimulates M1 microglia to differentiate into M2 microglia, eventually inducing protective microglial activities and Aβ clearance through micropinocytosis and phagocytosis. This limits AD progression [101,102,103,106]. In other words, Klotho overexpression can ameliorate Aβ deposition, synaptic loss and cognitive dysfunction through restraining the NLRP3/caspase 1 signaling pathway. This promotes the microglia transformation that clears Aβ.

### 3.3. Klotho Promote Aβ Transporter-Mediated Aβ Clearance

In addition to promoting microglia mediated Aβ clearance, an excessive expression of Klotho protein can also eliminate Aβ through regulating the expression of the Aβ transporter. It is known that Aβ is central to AD pathology; the relative levels and distribution of Aβ in the brain affect the development and progression of AD [7,15,34,107]. Continuous Aβ clearance in the CNS through the blood–brain barrier (BBB) and the blood–CSF barrier is important in preventing Aβ accumulation, thereby influencing disease progression. In the interstitial fluid (ISF), the concentration of Aβ is strongly regulated by the APP. The major transport mechanism of Aβ across the BBB is through different kinds of receptors. These include the transporters that transport Aβ into the brain, such as the receptor for advanced glycation end products (RAGE) and ATP-binding cassette transporter A1 (ABCA1). They also include transporters that transport Aβ out of the brain, such as low-density lipoprotein receptor-related protein-1 (LRP1) and P-glycoprotein (P-gp) [12,48,107,108,109]. The expression of Aβ transporter in the brain is important in regulating the specific receptor-mediated transport mechanism, which controls the circulation of Aβ that enters or diffuses out of the brain. Several findings have shown that when pathogenic Aβ deposits in the AD brain, the mRNA and protein levels of RAGE and ABCA1 are highly upregulated in the cerebral vessels and the brain of the transgenic mice model of β-amyloidosis, in which Aβ is facilitated to enter the brain via the BBB. At the same time, the mRNA and protein levels of LRP1 and P-gp are downregulated in the AD brain and cerebral vessels [35,107,110]. This reduces Aβ clearance and promotes Aβ deposition in the cerebrovascular tissue. It is important to notice that according to recent research the overexpression of the Klotho protein can facilitate Aβ transporter-mediated Aβ clearance by regulating the expression of Aβ-related transporters such as LRP1 and P-gp. Moreover, Klotho upregulates soluble LRP (sLRP) that has the potential to sequester extracellular Aβ, and promote its degradation in the periphery. With Klotho upregulation, the expression of the Aβ-efflux transporter is upregulated, and the Aβ-influx transporter is significantly attenuated in the cerebral vessels [35]. Thus, Aβ is rapidly cleared across the BBB, from the brain interstitial fluid into the blood, eventually, reducing inflammation and improving the recovery of cognitive function in the transgenic AD model.

### 3.4. Klotho Mitigates Tau Pathology and Enhances Cognitive Function in AD

The Tau protein stabilizes and maintains the integrity and structure of microtubules in the neuronal cells [15]. In AD, the Tau protein is hyperphosphorylated and aggregated intracellularly in the neurons, leading to neuronal damage and autophagy dysfunction [14]. Therefore, the increasing symptoms associated to Tau pathology in AD has raised a lot of questions to find new therapeutic targets which can subside the Tau pathogenesis and enhance cognitive function. Klotho has been reported to have close relation with aging related abnormalities and the autophagy pathway in AD [66]. Recent studies have reported that the regulation of the activity and expression level of the Klotho protein may be a potential therapeutic target against AD and tau pathology [51,73]. Tau protein has been reported to jointly work with kinesin adaptors for the transport of cargos, such as autophagosomes and lysosomes in the neurons during the anterograde and retrograde transport of cell organelles [53]. Therefore, the disruption of autophagy and autophagosome transport may be influenced by the hyperphosphorylated Tau in AD [111]. Thus, promoting or the induction of ALP in neurons may clear the production of Tau aggregates. A recent study has illustrated that Klotho regulates the induction of autophagy and ALP in neurons which supports the regulation of the intracellular process, and the Tau protein functions [34]. Another recent study illustrated that Klotho-VS heterozygosity in AD patients showed less Tau pathology symptoms and enhanced cognitive functions [112]. This study clearly demonstrated that neuroprotective effect of Klotho by decreasing Tau-related symptoms, and the PET imaging showed less intensity of Tau symptoms compared to Aβ pathology and reduced the cognitive impairment [112]. Taken together Klotho plays a concomitant role in the regulation of the intracellular process in neurons, and mitigates Tau pathogenesis by inducing autophagy and lysosome transport functions in AD pathology.

## 4. Klotho Acts as an Autophagy Inducer

### 4.1. Klotho Stimulates the Formation of UNC51-like Kinase-1 (ULK1) Complex

Autophagy is stimulated by AMP-activated protein kinase (AMPK), which is an energy sensor responsible for regulating cellular metabolism to maintain energy homeostasis [48,68]. In contrast, autophagy is suppressed by the p38 mitogen-activated protein kinase (MAPK) and the mammalian target of rapamycin (mTOR). mTOR is known as the major inhibitory signal of autophagy [11,47,57,113,114]. Extensive works have established that yeast Autophagy Related 1 Homolog (ATG1) kinase is a key regulator in autophagy stimulation; it complexes with ATG13 and ATG17 during autophagy. These proteins are needed for autophagy induction and autophagosome generation [48,57]. Mammalian homologs of ATG1 have been identified as Unc-51 Like Autophagy Activating Kinase 1 (ULK1) and Unc-51 Like Autophagy Activating Kinase 2 (ULK2) [50,57,58]. Mammalian counterparts of ATG13 and ATG17 are reported as mATG13 and a focal adhesion kinase family-interacting protein of 200 kDa (FIP200), respectively. ULK1 is a serine/threonine-protein kinase that plays a crucial role in autophagy induction. It interacts with FIP200 and ATG13 to form the ULK1/ATG13/FIP200 complex, which integrates autophagy signals into forming autophagosome. ATG13 and FIP200 are the autophagy-essential binding partner of ULK1. These proteins help regulate ULK1 kinase activity and correct its localization in the pre-autophagosome [46,50,57,62]. Recent work has shown that Klotho protein may have a role in autophagy induction by promoting the ULK1 complex formation [62]. As described previously, the initiation of autophagy is highly regulated by the ULK1-ATG13-FIP200 complex. It integrates the signals from upstream sensors, for example, p38 MAPK and mTOR, followed by the formation and degradation of the autophagosome. Autophagy induction is inhibited when the ULK1 complex formation is blocked. However, there is evidence that, under stress conditions, Klotho protein can stimulate the ULK1 complex formation and thus induce autophagy (Figure 4). Klotho silencing increases mTOR phosphorylation and is accompanied by the inactivation of AMPK. The activated mTOR attenuate ATG13 activation, thus, prevents the formation of the ULK1 complex and also inhibits the complex activity involved in autophagosome formation [45,61,62,113]. Therefore, those cells that could not induce the initial stage of autophagy were consequently accelerated to undergo apoptotic cell death [11,48,57,62]. Klotho protein may play a role in activating the ULK1/ATG13/FIP200 complex formation by mediating mTOR signaling [109,115], by activating ATG13, and by increasing ULK1 phosphorylation by AMPK [74,116]. As a consequence, the induction of the autophagy process suppresses the development and progression of AD by eliminating the APP plaques and inhibiting the neurofibrillary tangles formation [34,39,47].

### 4.2. Klotho Inhibits the IGF-1/PI3K/Akt/mTOR Pathway

Recent work has indicated that autophagy induction is a valid strategy to promote and sustain the survival of neuronal cells and clear abnormal protein aggregates from the brain [56,113]. It is acknowledged that autophagy disruption has a critical role in neurodegenerative disorders. Defects in the elimination of toxic and abnormal protein promotes cellular stress, and death [34,62,113]. The progression of autophagy involves a variety of signaling pathways. The PI3K/Akt/mTOR signaling pathway is one that has been widely examined. Insulin-like growth factors (IGF-1) which are mediated by the extracellular IGF-1 receptor (IGF-1R), function to enhance the survival and proliferation of specific tissues through activating the PI3K/Akt/mTOR signaling pathway [113,117]. In addition, the PI3K/Akt/mTOR signaling pathway takes a central role in maintaining cell viability, controlling cell growth and regulating autophagy [11,117,118,119]. PI3K/Akt is one of the upstream kinases that mediates the activity of mTOR. Studies have shown that the Akt and PI3K phosphorylation causes mTOR hyperactivation in AD patients [45]. It is known that PI3K/Akt is influenced by insulin or IGF-1, which is responsible for autophagy, actin cytoskeleton, and protein synthesis through the activity of mTOR [11,33]. It was reported that the inhibition of PI3K activity can remarkably suppress the downstream signaling of Akt and mTOR pathways. Downregulated mTOR activity stimulates autophagy by activating autophagy-related proteins, thus triggering the clearance of misfolded proteins [114,118]. Klotho influences IGF-1/PI3K/Akt/mTOR signaling regulation. Several studies have illustrated that Klotho can suppress the insulin/IGF-1 signaling pathway [26,27,61,69]. In vivo and in vitro assays have shown that the upregulation of Klotho results in an inhibition of IGF-1/PI3K/Akt/mTOR signaling, which suggests that the Klotho protein may affect PI3K/Akt/mTOR signaling regulation [34,43]. At the same time, it is proved that ligustilide (LIG)-induced Klotho overexpression can inhibit the insulin/IGF-1/PI3K/Akt signaling pathway, thereby suppressing the FoxO phosphorylation. Downregulated FoxO phosphorylation then results in the transcription of FoxO, consequently decreasing oxidative stress in the brain [27]. A Klotho deficiency is associated with a significant increase in IGF-1 phosphorylation, which enhances Akt and mTOR phosphorylation. It is likely that a deficiency of the Klotho protein would induce the enhancement of mTOR signaling, in turn, suppressing autophagy [39,43]. The upregulation of the Klotho protein has an effect on the suppression of the IGF-1/PI3K/Akt/mTOR signaling pathway, thereby facilitating Aβ clearance via sustained autophagy.

### 4.3. Klotho Promotes Nuclear Translocation of TFEB

Transcription factor EB (TFEB) belongs to the microphthalmia/transcription factor E (MiT/TFE) family [44,120,121,122,123,124]. There is abundant evidence that TFEB inhibits toxic and abnormal protein accumulation in AD cell and mouse models, and, in turn, improves behavioral deficits, cognitive impairment, and the mitigation of neurodegeneration [53,123,125]. TFEB regulates autophagy and lysosomal biogenesis by binding to the Coordinated Lysosomal Expression and Regulation (CLEAR) motif. Moreover, TFEB activation can regulate cellular metabolism, increase the number of lysosomes and induce lysosome-mediated degradation of toxic proteins by regulating the CLEAR genes [121,122,124,126,127]. For instance, some genes take part in the initiation of autophagy (BECN1, ATG9B and WIPI1), while others take part in autophagosome elongation (ATG5 and MAP1LC3B), substrate recognition, autophagosome docking and fusion with lysosomes (UVRAG and RAB7) [44,120,121,122]. The Klotho protein has been reported to have an effect on autophagy induction by promoting TFEB nuclear translocation and increasing TFEB-mediated lysosomal gene transcription [44,123]. mTORC1 and glycogen synthase kinase 3β (GSK3β) are classified as the kinases potentially responsible for TFEB phosphorylation in most of the cell-types, thereby preventing the nuclear translocation of TFEB [120]. One report has indicated that the Klotho protein has a role in inhibiting mTOR activity, as well as suppressing the phosphorylation of TFEB [39]. When TFEB is dephosphorylated, it quickly translocates into the nucleus, and promotes the transcription of genes necessary for autophagy and lysosomal biogenesis (Figure 5). To date, studies have also shown that Klotho promotes the inactivation of GSK3β, which in-turn leads to the dephosphorylation and nucleus localization of TFEB [32]. In an experimental mouse model, it was reported that mice treated with recombinant mouse Klotho (rKlotho) had an increased level of TFEB expression when compared to the wild type [44]. This suggests that Klotho can stimulate the nuclear translocation of TFEB, and TFEB translocation then promotes the lysosomal gene transcription. Hence, the lysosomal function improved by the Klotho protein facilitates the autophagy clearance of autophagosomes in a tacrolimus-induced renal injury mouse model [32,44]. Collectively, these findings highlight the role of the Klotho protein in improving autophagy clearance and lysosomal function by promoting TFEB translocation and increasing TFEB-mediated lysosomal gene transcription through inhibiting the kinase activity of mTORC1 and GSK3β.

## 5. Conclusions

Aging is a physiological change that occurs naturally, and mechanistically aging is developed by some main regulators, such as Klotho [128]. AD is developed due to the consequences of accelerated aging and some genetic mutational events with its own entity [128]. AD could be prevented or cured by targeting Klotho or can be achieved by healthy aging in a natural discipline [128]. Thus, developing therapeutic strategies to reduce cerebral Aβ deposition through repressing its generation and/or strengthening its clearance is critical for inhibiting AD progression [128]. Several lines of evidence support that Klotho may be a promising target for AD treatment. Its potential therapeutic value derives from its abilities to improve AD pathogenesis to reduce cognitive deficits and to stimulate autophagy. Klotho upregulation can improve Aβ clearance in an AD transgenic mice model, thereby alleviating cognition impairment and enhancing neuroprotection [128]. The addition of the Klotho protein can stimulate the ULK1 complex formation, TFEB nuclear translocation, as well as inhibit the IGF-1/PI3K/Akt/mTOR signaling pathway, which are all involved in autophagy activation. The mechanism of the Klotho protein involvement in AD has not been elucidated. Advancements in research on Klotho will undoubtedly bring a better understanding of its function and its mechanisms. As we have described above, accumulating evidence has shown that Klotho plays a neuroprotective function in AD. However, more deep mechanistic studies involving AD and autophagy remain elusive. The research reviewed here suggests that regulating the activity and expression level of the Klotho protein may be a potential therapeutic target against AD. Further studies are needed to explore this possibility of how Klotho is involved in autophagy processes. Moreover, other forms of autophagy and its clearance mechanisms in neurodegenerative diseases are still unknown. In future these questions need to be answered with many more research studies.

## Figures and Tables

**Figure 1 biomedicines-10-00705-f001:**
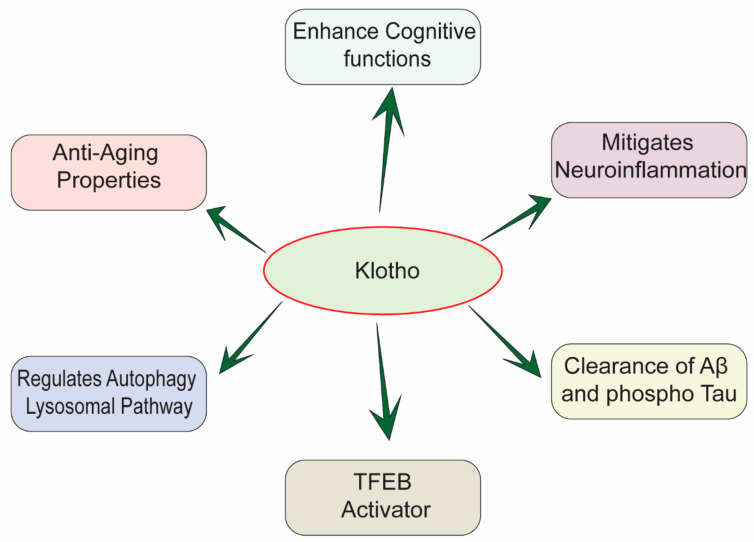
Multifactorial effects of Klotho in the pathology of AD.

**Figure 2 biomedicines-10-00705-f002:**
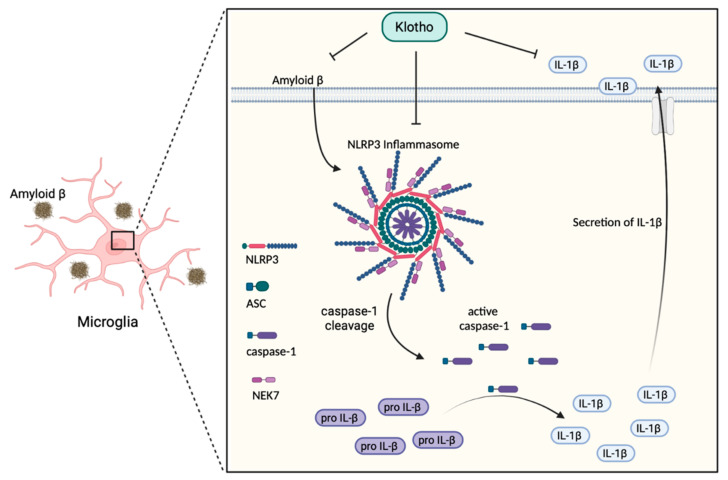
Effects of Klotho on the NLRP3/caspase-1 signaling pathway in AD. Klotho inhibits the accumulation of amyloid β peptide which can affect microglia directly and stimulate the NLRP3/caspase-1 signaling in the brain. The black arrow represents the consequence of Aβ stimulation. NLRP3: Nucleotide-binding oligomerization domain leucine-rich repeat and pyrin domain-containing protein 3; ASC: Apoptosis-associated speck-like protein containing a caspase recruitment domain; pro IL-1β: inactive interleukin-1β precursor; IL-1β: interleukin-1β.

**Figure 3 biomedicines-10-00705-f003:**
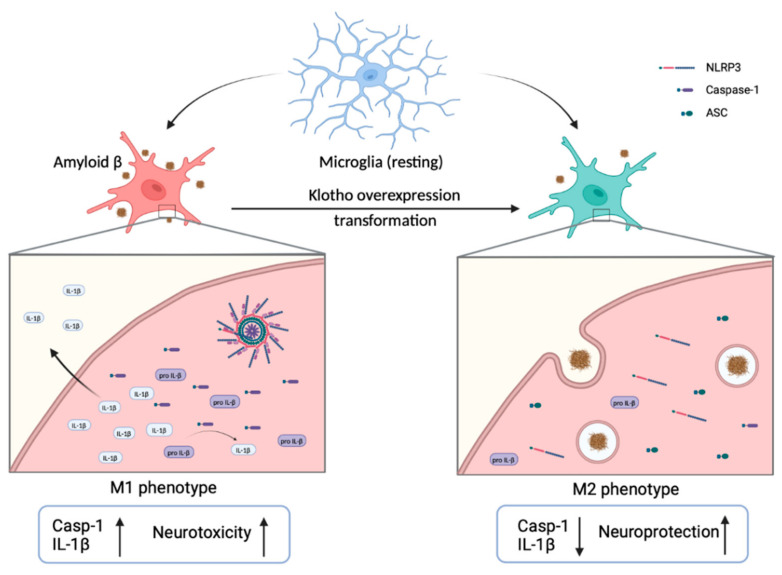
Transformation of microglia with the overexpression of Klotho. Microglia are usually in a resting state in the brain. Once activated, they show either M1 or M2 phenotypes. In the activated M1 phenotype, the expression of caspase-1 and IL-1β is enhanced, which stimulate a pro-inflammatory response and cause neurotoxicity. In contrast, when Klotho is overexpressed, M1 microglia are differentiated into M2 microglia. The secretion of caspase-1 and IL-1β is decreased, thereby facilitating Aβ clearance, preserving neurons and protecting against cognitive dysfunction.

**Figure 4 biomedicines-10-00705-f004:**
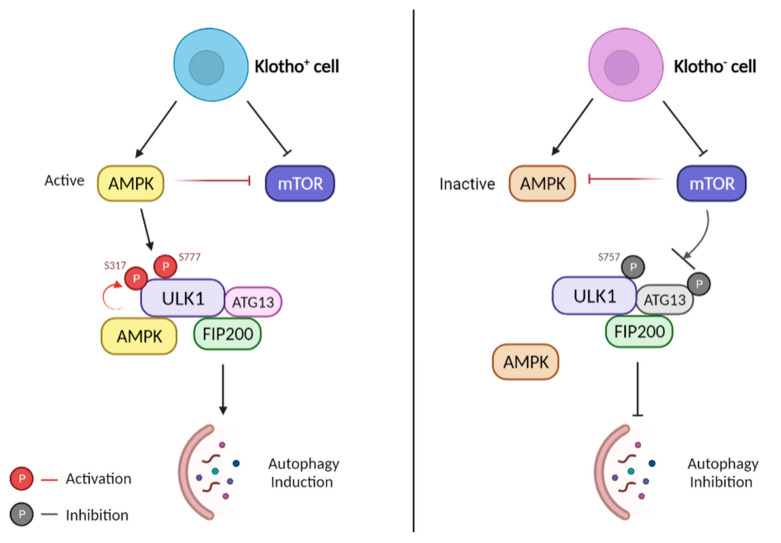
Role of Klotho protein in ULK1-induced autophagy. Left: Klotho overexpression. AMPK is active and mTOR is inhibited by AMPK and Klotho protein. ULK1 is subsequently phosphorylated by active AMPK at sites Ser 317 and 777. The complex of ULK1/ATG13/FIP200 is formed and autophagy is promoted. Right: Klotho deficiency. AMPK becomes inactive while mTOR becomes active. Activated mTOR phosphorylates ULK1 on Ser757, preventing ULK1 from interacting with AMPK, suppressing the formation of the ULK1 complex, and inhibiting autophagy.

**Figure 5 biomedicines-10-00705-f005:**
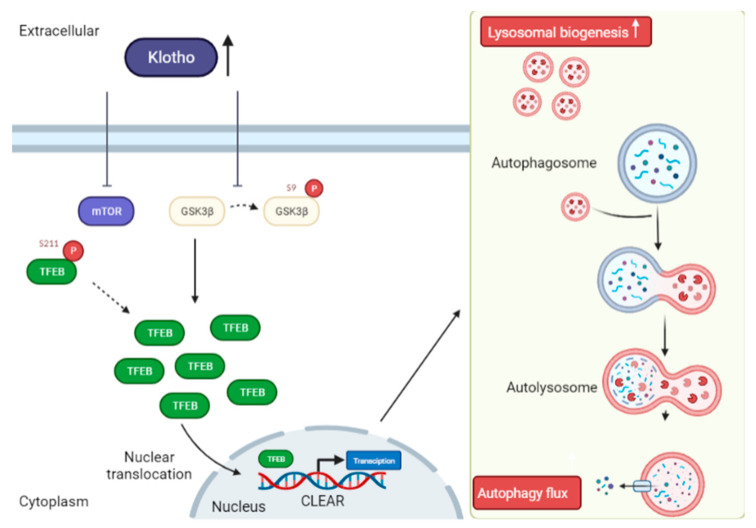
Mechanism of Klotho activity in autophagy induction through TFEB nuclear translocation. Soluble Klotho (sKlotho) or recombinant mouse Klotho (rKlotho) inhibits mTOR (S211) activity, thereby preventing TFEB phosphorylation resulting in nuclear translocation and activation of the transcription factor EB (TFEB). sKlotho or rKlotho also represses glycogen synthase kinase 3β (GSK3β) (S9) phosphorylation to trigger TFEB nuclear translocation. Improved TFEB-mediated lysosomal gene transcription by sKlotho increases lysosomal biogenesis and eventually enhances autophagy flux.

**Table 1 biomedicines-10-00705-t001:** Recent works on Klotho protein in Alzheimer’s disease (AD).

Author, Year	Title	Finding	Reference
Kuang et al., 2014	Klotho upregulation contributes to the neuroprotection of ligustilide in an Alzheimer’s disease mouse model	Ligustilide (LIG)-induced Klotho overexpression is neuroprotective towards AD by downregulating the insulin/IGF-1 signaling pathway, thereby triggering Forkhead-box class O (FoxO) transcription factor to relieve oxidative stress in the brain. Ligustilide increased mitochondrial manganese-superoxide dismutase, catalase expression and activity, and decreased malondialdehyde, protein carbonyl, and 8-hydroxydesoxyguanosine levels in the brain.	[27]
Kuang et al., 2017	Neuroprotective effect of Ligustilide through induction of α-secretase processing of both APP and Klotho in a mouse model of Alzheimer’s disease	Ligustilide (LIG)-induced the expression of both soluble APPα (sAPPα) and soluble Klotho (sKL) protein, thus facilitating the inhibition of IGF-1/Akt/mTOR signaling. The neuroprotective role of LIG against AD is highly associated with an increased level of Klotho, and ADAM10 proteins, eventually, promoting cerebral Aβ clearance and improving cognitive function.	[43]
Zeng et al., 2019	Lentiviral vector mediated overexpression of Klotho in the brain improves Alzheimer’s disease-like pathology and cognitive deficits in mice	Klotho protein is strongly associated with Aβ clearance via AKT/mTOR signaling pathway repression and the activation of the autophagy-lysosome system, resulting in the improvement of cognitive function and beneficial pathological changes in an AD mouse model.	[34]
Zhao et al., 2020	Klotho overexpression improves amyloid-β clearance and cognition in the APP/PS1 mouse model of Alzheimer’s disease	Klotho treatment remarkably improved AD-induced neuropathies in an aged transgenic mice model. It suppressed NLRP3, promoted microglia transformation and modulated the expression of Aβ transporter, eventually enhancing Aβ clearance in the brain.	[35]

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
