# Peer review of "Klotho an Autophagy Stimulator as a Potential Therapeutic Target for Alzheimer’s Disease: A Review"

_biomedicines, 2022, doi:10.3390/biomedicines10030705_

Round 1

Reviewer 1 Report

In the manuscript entitled “ Klotho an autophagy stimulator as a potential therapeutic tar-2 get for Alzheimer’s disease: A review”, the authors who have extensive experience in this research topic will try to collect information on a novel β-glucuronidase family member, Klotho in terms of its participation in the physiology and pathophysiology of the nervous system. The article is very interesting and well written both in terms of content and form. However, sometimes some linguistic errors do occur (for example, line 96, something is missing from this sentence). In order to make the text clearer, it would also be worthwhile to use paragraph division more often.

Describing the activities of Klotho, the authors showed very well how important it is for the support of life expectancy, for the regulation of autophagy, and for the maintenance of the balance of inflammatory processes in the nervous system.

Below are some considerations to be consider:

  • The authors thoroughly describe the action of Klotho regulating the metabolism of amyloid and supporting the removal of its toxic forms.However, there is no information, apart from a brief mention in the Introduction, about the role of Klotho in the regulation of the physiology of tau, the second key protein undergoing pathological changes in neurodegenerative diseases.

In case of the distal parts of axons, autophagosomes are produced there and transported to the soma in a retrograde fashion, where they fuse with lysosomes, forming autolysosomes in which the degradation of their content is completed. More recently, it has been shown that soma-derived degradative lysosomes are transported in an anterograde way from soma into distal axons, where they fuse with autophagosomes, forming degradative autolysosomes, some of which move back to soma. By knocking down Arl8, a lysosomal kinesin adaptor, the delivery of lysosomes to axonal terminals was impaired, and an aberrant accumulation of autophagosomes in distal part of axons was observed. Both anterograde and to a lesser extent retrograde transport along microtubules is influenced by microtubule-associated tau thus, it is likely that autophagosome transport in neurites may be influenced, also adversely, by tau. Disrupted autophagy led to the formation of TauOs and insoluble aggregates. Induction or enhancement of the autophagy process may limit the production of these tau moieties. Taken together, these findings raise the likely possibility of involvement of Klotho in the regulation of intracellular processes dependent on the proper tau function.

  • The authors suggest that the regulation of the activity and expression level of Klotho protein may be a potential therapeutic target against AD. Therefore, it would be worth including a chapter discussing the results of studies in humans with the KL-VShet heterozygous genotype. Some findings revealed a protective association of KL-VShet on tau accumulation that particularly manifested in amyloid-positive individuals, where lower tau pathology was related to better cognitive functions. Elderly KL-VShet carriers with elevated Aβ burden, i.e., the earliest primary AD pathology, exhibited lower tau-PET levels and tau-PET annual change rates when compared to those in KL-VShet non-carriers. These findings may be particularly informative for clinical anti-tau trials and may encourage future studies on enhancing Klotho protein levels as a therapeutic intervention to slow down the development of tau pathology and dementia in AD.

  • There have also been interventional studies inducing increased expression of Klotho. The viral vector-mediated overexpression of Klotho in the brain improves Alzheimer’s disease like pathology and cognitive deficits in murine model of AD (C.-Y. Zeng et al. / Neurobiology of Aging 78 (2019) and a trial of human clinical study (J Regen Biol Med. 2021; 3(6):1-15). A hint of this research would be in line with the profile of the journal titled “Biomedicines”.

Author Response

Point-by-point rebuttal letter to the expert reviewers

Reviewer-1

In the manuscript entitled “Klotho an autophagy stimulator as a potential therapeutic tar-get for Alzheimer’s disease: A review”, the authors who have extensive experience in this research topic will try to collect information on a novel β-glucuronidase family member, Klotho in terms of its participation in the physiology and pathophysiology of the nervous system. The article is very interesting and well written both in terms of content and form. However, sometimes some linguistic errors do occur (for example, line 96, something is missing from this sentence). In order to make the text clearer, it would also be worthwhile to use paragraph division more often.

Response: Thank you for your comment. We have re-framed the sentence to convey the message clearly.

Describing the activities of Klotho, the authors showed very well how important it is for the support of life expectancy, for the regulation of autophagy, and for the maintenance of the balance of inflammatory processes in the nervous system.

Response: Thank you for your comment.

Below are some considerations to be consider:

The authors thoroughly describe the action of Klotho regulating the metabolism of amyloid and supporting the removal of its toxic forms. However, there is no information, apart from a brief mention in the Introduction, about the role of Klotho in the regulation of the physiology of tau, the second key protein undergoing pathological changes in neurodegenerative diseases.

Response: Thank you for your comment. We have added the necessary information in section 3.4. Klotho mitigates Tau pathology and enhance cognitive function in AD in the manuscript.

In case of the distal parts of axons, autophagosomes are produced there and transported to the soma in a retrograde fashion, where they fuse with lysosomes, forming autolysosomes in which the degradation of their content is completed. More recently, it has been shown that soma-derived degradative lysosomes are transported in an anterograde way from soma into distal axons, where they fuse with autophagosomes, forming degradative autolysosomes, some of which move back to soma. By knocking down Arl8, a lysosomal kinesin adaptor, the delivery of lysosomes to axonal terminals was impaired, and an aberrant accumulation of autophagosomes in distal part of axons was observed. Both anterograde and to a lesser extent retrograde transport along microtubules is influenced by microtubule-associated tau thus, it is likely that autophagosome transport in neurites may be influenced, also adversely, by tau. Disrupted autophagy led to the formation of TauOs and insoluble aggregates. Induction or enhancement of the autophagy process may limit the production of these tau moieties. Taken together, these findings raise the likely possibility of involvement of Klotho in the regulation of intracellular processes dependent on the proper tau function.

The authors suggest that the regulation of the activity and expression level of Klotho protein may be a potential therapeutic target against AD. Therefore, it would be worth including a chapter discussing the results of studies in humans with the KL-VShet heterozygous genotype. Some findings revealed a protective association of KL-VShet on tau accumulation that particularly manifested in amyloid-positive individuals, where lower tau pathology was related to better cognitive functions. Elderly KL-VShet carriers with elevated Aβ burden, i.e., the earliest primary AD pathology, exhibited lower tau-PET levels and tau-PET annual change rates when compared to those in KL-VShet non-carriers. These findings may be particularly informative for clinical anti-tau trials and may encourage future studies on enhancing Klotho protein levels as a therapeutic intervention to slow down the development of tau pathology and dementia in AD.

There have also been interventional studies inducing increased expression of Klotho. The viral vector-mediated overexpression of Klotho in the brain improves Alzheimer’s disease like pathology and cognitive deficits in murine model of AD (C.-Y. Zeng et al. / Neurobiology of Aging 78 (2019) and a trial of human clinical study (J Regen Biol Med. 2021; 3(6):1-15). A hint of this research would be in line with the profile of the journal titled “Biomedicines”.

Response: We thank the reviewer for their comment. In Section 3.1 (line 348), we have mentioned that in aged APP/PS1 mice, viral vector-mediated overexpression of Klotho significantly reduced phosphorylated tau and inhibited tau pathology. Furthermore, we have incorporated the valuable information provided by the reviewer.

We have added the necessary information in the manuscript with a subtitled section 3.4. We have included all the reference mentioned by the reviewers.

Reviewer 2 Report

In the INTRODUCTION section (line 58), the authors say that it has recently been related to various processes. It would be preferable to remove the word “recently”, because some of them were published for almost a decade (Nagai T, et al. FASEB J. 2003; Xu ZL, et al. Acta Pharmacol Sin. 2004; Mitobe M, et al. Nephron Exp Nephrol. 2005; etc).

In section 2 (Expression, structure and function of Klotho protein), a curious fact that could be added in line 111 (talking about Greek mythology) is that Klotho (Clotho) was the youngest daughter of Zeus and she (“spinner” ) spun the thread of life from her distaff onto her spindle. [That’s why they gave the name to this gene and this protein, related to longevity].

Also in this section, they could introduce some sentence about the anti-fibrosing role that Klotho has. This idea can be important for the conclusions.

Regarding this section of conclusions, it could be interesting to highlight in some way that for a better approximation to the therapeutic strategies of AD, we should remember that this disease is not only the consequence of an accelerated aging process and that therefore it could be prevent or cure through healthy aging. Beyond this perspective, it should be emphasized that AD is a condition with its own entity. To focus this idea, we recommend that authors consider the article published by Nelson PT, et al (Acta Neuropathol. 2011. Alzheimer disease is not “brain aging”: neuropathological, genetic, and epidemiological human studies.)

In other words, it could be that physiological aging is related to a small part of the mechanisms in which Klotho has some role as an inhibitor (such as oxidative stress and fibrosis, which participate in the atherogenesis process and, therefore, in the vascular dementia). However, as the authors have masterfully exposed, Klotho has crucial actions in the etiopathogenesis of Alzheimer. Therefore, the key in the search for therapies could be conduct the research towards these specific mechanisms that Klotho has in AD (and not to the processes of aging in general).

Author Response

Point-by-point rebuttal letter to the expert reviewers

Reviewer-2

In the INTRODUCTION section (line 58), the authors say that it has recently been related to various processes. It would be preferable to remove the word “recently”, because some of them were published for almost a decade (Nagai T, et al. FASEB J. 2003; Xu ZL, et al. Acta Pharmacol Sin. 2004; Mitobe M, et al. Nephron Exp Nephrol. 2005; etc).

Response: Thank you for your comment. We have replaced “recently” to “previous” in line 62 of the revised manuscript.

In section 2 (Expression, structure and function of Klotho protein), a curious fact that could be added in line 111 (talking about Greek mythology) is that Klotho (Clotho) was the youngest daughter of Zeus and she (“spinner” ) spun the thread of life from her distaff onto her spindle. [That’s why they gave the name to this gene and this protein, related to longevity].

Response: Thank you for your comment. We have added the necessary information in line 179 of the revised manuscript.

Also in this section, they could introduce some sentence about the anti-fibrosing role that Klotho has. This idea can be important for the conclusions.

Response: Thank you for your comment. We have added the necessary information in line 260-265 of the revised manuscript.

Regarding this section of conclusions, it could be interesting to highlight in some way that for a better approximation to the therapeutic strategies of AD, we should remember that this disease is not only the consequence of an accelerated aging process and that therefore it could be prevent or cure through healthy aging. Beyond this perspective, it should be emphasized that AD is a condition with its own entity. To focus this idea, we recommend that authors consider the article published by Nelson PT, et al (Acta Neuropathol. 2011. Alzheimer disease is not “brain aging”: neuropathological, genetic, and epidemiological human studies.)

Response: Thank you for your valuable suggestion. As suggested, we have added the reference article and have changed the related information in the conclusion section of the revised manuscript. 

In other words, it could be that physiological aging is related to a small part of the mechanisms in which Klotho has some role as an inhibitor (such as oxidative stress and fibrosis, which participate in the atherogenesis process and, therefore, in the vascular dementia). However, as the authors have masterfully exposed, Klotho has crucial actions in the etiopathogenesis of Alzheimer. Therefore, the key in the search for therapies could be conduct the research towards these specific mechanisms that Klotho has in AD (and not to the processes of aging in general).

Response: Thank you for your valuable suggestion. As suggested, we have added the reference article and have changed the related information in the conclusion section.

Reviewer 3 Report

Comments to klotho review 24Feb2022

The review paper intends to describe the effect Klotho has on various intracellular processes that ultimately affect the development of Alzheimer’s disease. The topic is interesting, but the text is unorganized and superficial. Also, there are several scientific inaccuracies in the text that need to be corrected. The english needs to be revised by a scientific person to avoid inaccurate sentences. The title provides an impression that Klotho is a potential target – but the text lacks data showing how it can be targeted. Actually – increased expression of Klotho should lead to improvement of the disease. There is a lack of information how Klotho causes all of the mentioned effects when it is a transmembrane protein. Could it be that the soluble Klotho is responsible for the mentioned functions? It means that the sheddases are lower in Alzheimer’s disease?

The abstract should be more concise and describe more the main findings (the following sentence can be removed: “A review and summary of the 21 current understanding of how Klotho regulates and affects the development of AD may be helpful”).

Please provide a subtitle to the text that appears immediately after Table 1. Also make some more subtitles that will help to build up a more structured flow of the text. In the current form, there are many jumps. More detailed information is required.

Please also provide some more mechanistic insight – how does Klotho affect the various cell processes mentioned in the text in light of the fact that it is a transmembrae protein. What are the natural ligands of Klotho?

In Table 1: One place the APP is in small letters. Please use capital letters.

Please add to the table the data that Ligustilide increased mitochondrial manganese-superoxide dismutase and catalase expression and activity, and decreased malondialdehyde, protein carbonyl, and 8-hydroxydesoxyguanosine levels in the brain.

Also a mistake in the table: It says that “LIG induced the overexpression of APP” – the intention would be ‘induced the alpha-processing of APP, thus increasing the level of soluble APP”.

Also important to mention the upregulation of ADAM10.

Also describe in more detail how Ligustilide inhibits the IGF-1/Akt/mTOR signaling (that is important to cell survival) and causes neuroprotection through upregulation of Forkhead-box class O (FoxO) transcription factors which are known to promote apoptosis.

Does Ligustilide have additional activities?

Line 69: The word “mediating” does not seem to be the right world. How can it mediate the Aβ-related transporters and microglia transformation?

Line 96: The world however  is not proper here. “However” refers to a contrast appearing in previous sentence. But here is the first time Klotho appears. Also the sentence “other studies have Klotho protein can stimulate” has a syntactic error.

Line 97: “If the latter is true” should be removed. As if you do not believe in what you wrote in previous sentence that have 9 references.

Thee original papers showing an interaction between Klotho and FGFRs should be cited in lines 127-128.

Line 105 – remove novel. It is already known from 1997.

Line 129: It can not upregulate their binding to FGF23 but increasing their affinity to FGF23.

What is known about FGF23 and Alzheimer’s disease?

The sentences of lines 132-134 should have references.

Line 134: Rephrase “increasing studies”.

Line 143: spelling mistake. Correct to vanilloid.

Line 181: The title of Section 3 should be rephrased to what you are telling in the section.

Line 233: correct to “are prominent…”

Line 233:Instead of “indicates” write “suggests”.

Line 272: Correct to “promotes”

Line 296 – Please mention which Abeta efflux transporters are upregulated, and how Klotho causes this upregulation.

A figure summarizing Section 3 ought to be added.

Line 367 – please write the full name of LIG.

Line 403: add clearance of what.

Line 403 – there is a hyphen on the paranthesis. Please remove.

Please refer to the figures in the text.  

Figure 1 should include Klotho.

Figure 3 has a problem with the arrows and inhibitory signs in the Klotho negative cells.  Also at the bottom it should be “Autophagy inhibition”.

In Figure 4 Klotho is shown as an extracellular protein. Please state in the legend that it is the soluble form of Klotho.

A figure showig the multiple effects of Klotho as described in the Introduction would be helpful.

Many of the references are incomplete (lacking page number etc). (e.g., references 6, 9, 12, 13, 19, 25, 27, 35, 40, 55, 58, 66, 68, 69, 71, 73, 77, 78, 79, 82, 97).

The papers PMID: 34737707 and 35180928 should be mentioned, and please state what you renew over this review paper.

Author Response

Point-by-point rebuttal letter to the expert reviewers

Reviewer-3

The review paper intends to describe the effect Klotho has on various intracellular processes that ultimately affect the development of Alzheimer’s disease. The topic is interesting, but the text is unorganized and superficial. Also, there are several scientific inaccuracies in the text that need to be corrected. The english needs to be revised by a scientific person to avoid inaccurate sentences. The title provides an impression that Klotho is a potential target – but the text lacks data showing how it can be targeted. Actually – increased expression of Klotho should lead to improvement of the disease. There is a lack of information how Klotho causes all of the mentioned effects when it is a transmembrane protein. Could it be that the soluble Klotho is responsible for the mentioned functions? It means that the sheddases are lower in Alzheimer’s disease?

Response: We thank the reviewer for their comments. The English correction and scientific inaccuracies in the manuscript have been revised by a scientific expert in the field of neurodegenerative disease, including Alzheimer’s disease.

Klotho is a 130 kDa single span, transmembrane protein, also known as α-Klotho. Klotho protein is majorly comprises of three forms namely, full-length transmembrane form (mKl), secreted form (sKl), and secreted truncated form (KL1). According to earlier reports, the major sheddases that cleave mKl are ADAM10, and ADAM17. Juxta-membrane cleavage of mKl releases sKl, which possess several biological functions in distant organs, including anti-ageing, anti-fibrotic, cardioprotection, and neuroprotection.

Loss of function in ADAM-17, and alpha-secretase leads to the accumulation of Aβ and decrease in sheddases has been reported in AD [72,73].

The abstract should be more concise and describe more the main findings (the following sentence can be removed: “A review and summary of the 21 current understanding of how Klotho regulates and affects the development of AD may be helpful”).

Response: Thank you for your comment. After considering the reviewer’s suggestion, we have removed the sentence in line 21 of the manuscript.

Please provide a subtitle to the text that appears immediately after Table 1. Also make some more subtitles that will help to build up a more structured flow of the text. In the current form, there are many jumps. More detailed information is required.

Response: Thank you for your comment. After considering the reviewer’s suggestion, we have added some more information for klotho and autophagy.

Disruption of the beclin 1-BCL2 autophagy regulatory complex promotes longevity in mice. Nature. 2018 Jun;558(7708):136-140. doi: 10.1038/s41586-018-0162-7. Epub 2018 May 30.

Please also provide some more mechanistic insight – how does Klotho affect the various cell processes mentioned in the text in light of the fact that it is a transmembrae protein. What are the natural ligands of Klotho?

Response: We thank the reviewer for their comments. We have provided the necessary information in the revised manuscript.

Recent evidence strongly suggest that enhancing the expression of Klotho is a promising strategy to counter age-related disorders including Alzheimer’s disease. In this study, administration of senescent cells decreased the expression of urinary, kidney, and brain-specific Klotho, via the production of senescence-associated secretory phenotype (SASP) factors in old and obese mice. This effect was significantly rescued after treating the mice with senolytic compounds such as Dasatinib+Quercetin (D+Q) or Fisetin.

Yi Zhu, Larissa G.P. Langhi Prata, Erin O. Wissler Gerdes, Jair Machado Espindola Netto, Tamar Pirtskhalava, Nino Giorgadze, Utkarsh Tripathi, Christina L. Inman, Kurt O. Johnson, Ailing Xue, Allyson K. Palmer, Tingjun Chen, Kalli Schaefer, Jamie N. Justice, Anoop M. Nambiar, Nicolas Musi, Stephen B. Kritchevsky, Jun Chen, Sundeep Khosla, Diana Jurk, Marissa J. Schafer, Tamar Tchkonia, James L. Kirkland. Orally-active, clinically-translatable senolytics restore α-Klotho in mice and humans. eBioMedicine, 2022, 103912, ISSN 2352-3964, https://doi.org/10.1016/j.ebiom.2022.103912.

In Table 1: One place the APP is in small letters. Please use capital letters.

Response: Thank you for your comment. We have changed “app” to “APP” in Table 1 of the manuscript.

Please add to the table the data that Ligustilide increased mitochondrial manganese-superoxide dismutase and catalase expression and activity, and decreased malondialdehyde, protein carbonyl, and 8-hydroxydesoxyguanosine levels in the brain.

Response: Thank you for your valuable suggestion. As suggested, we have added the data into the table

Also a mistake in the table: It says that “LIG induced the overexpression of APP” – the intention would be ‘induced the alpha-processing of APP, thus increasing the level of soluble APP”.

Response: We thank the reviewer for carefully checking. We have added “soluble APPα (sAPPα) and soluble Klotho (sKL)” in Table 1.

Also important to mention the upregulation of ADAM10.

Response: We thank the reviewer for their suggestion. We have added “ADAM10” in Table 1.

Also describe in more detail how Ligustilide inhibits the IGF-1/Akt/mTOR signaling (that is important to cell survival) and causes neuroprotection through upregulation of Forkhead-box class O (FoxO) transcription factors which are known to promote apoptosis.Does Ligustilide have additional activities?

Response: We thank the reviewer for their suggestion. We have included as per your comments.

Line 69: The word “mediating” does not seem to be the right world. How can it mediate the Aβ-related transporters and microglia transformation?

Response: We thank the reviewer for their comment. We have replaced “through mediating” to “by regulating” in line 72 of the revised manuscript.

Line 96: The world however  is not proper here. “However” refers to a contrast appearing in previous sentence. But here is the first time Klotho appears. Also the sentence “other studies have Klotho protein can stimulate” has a syntactic error.

Response: We thank the reviewer’s comment. We agree with the reviewer and  “However” have been replaced with “Also” in line 124 of the manuscript. We have included ‘other studies have shown that Klotho protein can stimulate’ in line 125 of the revised manuscript.

Line 97: “If the latter is true” should be removed. As if you do not believe in what you wrote in previous sentence that have 9 references.

Response: We thank the reviewer for their comment. Since “If the latter is true” indeed contradicts the statements previously mentioned, we have replaced it with “Therefore, based on previous evidences”.

Thee original papers showing an interaction between Klotho and FGFRs should be cited in lines 127-128.

Response: We thank the reviewer for their suggestion. We have replaced the previous references to original papers in line 203 as shown below,

  1. Christian Lerch, Rukshana Shroff, Mandy Wan, Lesley Rees, Helen Aitkenhead, Ipek Kaplan Bulut, Daniela Thurn, Aysun Karabay Bayazit, Anna Niemirska, Nur Canpolat, Ali Duzova, Karolis Azukaitis, Ebru Yilmaz, Fatos Yalcinkaya, Jerome Harambat, Aysel Kiyak, Harika Alpay, Sandra Habbig, Ariane Zaloszyc, Oguz Soylemezoglu, Cengiz Candan, Alejandra Rosales, Anette Melk, Uwe Querfeld, Maren Leifheit-Nestler, Anja Sander, Franz Schaefer, Dieter Haffner, 4C study consortium, ESPN CKD-MBD working group, Effects of nutritional vitamin D supplementation on markers of bone and mineral metabolism in children with chronic kidney disease, Nephrology Dialysis Transplantation, Volume 33, Issue 12, December 2018, Pages 2208–2217, https://doi.org/10.1093/ndt/gfy012.

  1. Yuan, Q., Sitara, D., Sato, T., Densmore, M., Saito, H., Schüler, C., Erben, R. G., & Lanske, B. (2011). PTH ablation ameliorates the anomalies of Fgf23-deficient mice by suppressing the elevated vitamin D and calcium levels.Endocrinology,152(11), 4053–4061. https://doi.org/10.1210/en.2011-1113.

  1. Kolek OI, Hines ER, Jones MD, LeSueur LK, Lipko MA, Kiela PR, Collins JF, Haussler MR, Ghishan FK. 1alpha,25-Dihydroxyvitamin D3 upregulates FGF23 gene expression in bone: the final link in a renal-gastrointestinal-skeletal axis that controls phosphate transport. Am J Physiol Gastrointest Liver Physiol. 2005 Dec;289(6):G1036-42. doi: 10.1152/ajpgi.00243.2005. Epub 2005 Jul 14. PMID: 16020653.

Line 105 – remove novel. It is already known from 1997.

Response: We thank the reviewer for their comment. We have removed “novel” in line 173 pf the manuscript.

Line 129: It can not upregulate their binding to FGF23 but increasing their affinity to FGF23.

Response: We thank the reviewer for their suggestion. We have replaced “upregulates their binding” to “has an increased affinity towards” in line 200-201 in the revised manuscript.

What is known about FGF23 and Alzheimer’s disease?

Response: We thank the reviewer for their suggestion. We have included as per your comments in the line 149 to 152.

The sentences of lines 132-134 should have references.

Response: We thank the reviewer for their suggestion. We have included as per your comments.

Line 134: Rephrase “increasing studies”.

Response: We thank the reviewer’s comment. We have replaced “increasing studies” to “Emerging evidences” in line 213 of the manuscript.

Line 143: spelling mistake. Correct to vanilloid.

Response: We thank the reviewer’s comment. We have corrected the spelling in line 226 of the manuscript.

Line 181: The title of Section 3 should be rephrased to what you are telling in the section.

Response: We thank the reviewer’s comment. We have changed the title of Section 3 to “Klotho inhibits neuroinflammation, promotes Aβ clearance, and mitigates tau pathology in Alzheimer’s disease” in line 312 of the manuscript.

Line 233: correct to “are prominent…”

Response: We thank the reviewer’s comment. We have changed “is one of the” to “are” in line 379 of the manuscript.

Line 233:Instead of “indicates” write “suggests”.

Response: We thank the reviewer’s comment. We have changed “indicates” to “suggests” in line 379 of the manuscript.

Line 272: Correct to “promotes”

Response: We thank the reviewer for their suggestion. We have corrected the spelling to “promote” in line 426 of the manuscript.

Line 296 – Please mention which Abeta efflux transporters are upregulated, and how Klotho causes this upregulation.

Response: We thank the reviewer’s comment. According to previous study, lentivirus-mediated upregulation of Klotho, significantly enhanced the mRNA and protein levels of Aβ-efflux transporters such as LRP1 and P-gp in APP/PS1 mice. Moreover, Klotho upregulates soluble LRP (sLRP) that has the potential to sequester extracellular Aβ and promote its degradation in the periphery. We have added the necessary information in line 459 of the manuscript.

A figure summarizing Section 3 ought to be added.

Response: We thank the reviewer for their suggestion. We have included as per your comments.

Line 367 – please write the full name of LIG.

Response: Thank you for your comment. We have given the full name of LIG in line 580 of the revised manuscript.

Line 403: add clearance of what.

Response: We thank the reviewer for carefully noticing the incomplete sentence. We have added the necessary information in line 782-783 of the revised manuscript.

Line 403 – there is a hyphen on the paranthesis. Please remove.

Response: We thank the reviewer for their comment. We have removed the hyphen on the parenthesis in line 782 of the revised manuscript.

Please refer to the figures in the text.  

Response: We thank the reviewer for their suggestion. We have included as per your comments.

Figure 1 should include Klotho.

Response: We thank the reviewer for their suggestion. We have included as per your comments.

Figure 3 has a problem with the arrows and inhibitory signs in the Klotho negative cells.  Also at the bottom it should be “Autophagy inhibition”.

Response: We thank the reviewer for their suggestion. We have changed the figure as per your comments.

In Figure 4 Klotho is shown as an extracellular protein. Please state in the legend that it is the soluble form of Klotho.

Response: We thank the reviewer for their suggestion. In the figure legends of Figure 5, we have changed “Klotho” to “soluble Klotho (sKlotho)” in line 789 of the revised manuscript.

A figure showig the multiple effects of Klotho as described in the Introduction would be helpful.

Response: We thank the reviewer for their suggestion. We have included as per your comments as figure 1 at the end of introduction.

Many of the references are incomplete (lacking page number etc). (e.g., references 6, 9, 12, 13, 19, 25, 27, 35, 40, 55, 58, 66, 68, 69, 71, 73, 77, 78, 79, 82, 97).

Response: We thank the reviewer for their suggestion. We have made the necessary changes in the reference section of the revised manuscript.

The papers PMID: 34737707 and 35180928 should be mentioned, and please state what you renew over this review paper.

Response: We thank the reviewer for their suggestion. We have added the necessary references in the revised manuscript.

Round 2

Reviewer 3 Report

The manuscript has been significantly improved. There are still several places where the English needs to be improved. An Abbreviation list can be added at the end of the text.

In Figure legend 2 – Please add text on the effects of Klotho.

Line 399: Correct “ULK1 subsequently can phosphorylate by” to “ULK1 is subsequently phosphorylated by”

Line 466: I think rKlotho should be sKlotho (for soluble Klotho).

In Figure 5: The Klotho in the extracellular compartment should be written sKlotho.

Figure legend 5: The following sentence is not clear: “which stimulates transcription factor EB (TFEB) nuclear translocation, which suppresses TFEB phosphorylation.” I think you mean “thereby preventing TFEB phosphorylation resulting in nuclear translocation and activation of the transcription factor EB (TFEB)”.

Author Response

Point-by-point rebuttal letter to the expert reviewers

Reviewer-3

The manuscript has been significantly improved. There are still several places where the English needs to be improved. An Abbreviation list can be added at the end of the text.

Response: We thank the reviewer for their comments. We have made the necessary changes, and we have added the abbreviation section at the end of the conclusion.

In Figure legend 2 – Please add text on the effects of Klotho.

Response: We thank the reviewer for their comments. We have changed the figure legend as per the suggestion.

Line 399: Correct “ULK1 subsequently can phosphorylate by” to “ULK1 is subsequently phosphorylated by”

Response: Thank you for the comments. We have changed the figure legend as per the reviewer’s suggestion in line 413 of the revised manuscript.

Line 466: I think rKlotho should be sKlotho (for soluble Klotho).

Response: We thank the reviewer for their comments. We have changed it to “recombinant mouse Klotho (rKlotho)” in line 481 in accordance with the cited article [44] in the revised manuscript.

In Figure 5: The Klotho in the extracellular compartment should be written sKlotho.

Response: We thank the reviewer for their comments. As per several publications, recombinant mouse Klotho (rKlotho) and soluble Klotho (sKlotho) has been used in the text. Therefore, we have used Klotho as a common term in figure 5.

Figure legend 5: The following sentence is not clear: “which stimulates transcription factor EB (TFEB) nuclear translocation, which suppresses TFEB phosphorylation.” I think you mean “thereby preventing TFEB phosphorylation resulting in nuclear translocation and activation of the transcription factor EB (TFEB)”.

Response: Thank you for the comments. We have made the changes as per the reviewer’s suggestion.
